# Research Progress on the Application of Neutralizing Nanobodies in the Prevention and Treatment of Viral Infections

**DOI:** 10.3390/microorganisms13061352

**Published:** 2025-06-11

**Authors:** Qingling Duan, Tong Ai, Yingying Ma, Ruoyu Li, Hanlin Jin, Xingyi Chen, Rui Zhang, Kunlu Bao, Qi Chen

**Affiliations:** 1National Engineering Research Center for Bioengineering Drugs and the Technologies, Jiangxi Provincial Key Laboratory of Bioengineering Drugs, Institute of Translational Medicine, Jiangxi Medical College, Nanchang University, Nanchang 330031, China; dqldoctor@163.com (Q.D.); 4209122006@ncu.edu.cn (T.A.); 4207122100@ncu.edu.cn (Y.M.); 15717001503@163.com (R.L.); 4217123175@ncu.edu.cn (H.J.); 5701123295@ncu.edu.cn (X.C.); 15079133918@163.com (R.Z.); 2Joint Programme of Nanchang University and Queen Mary University of London, Jiangxi Medicine School, Nanchang University, Nanchang 330031, China; 3Department of Developmental BioEngineering, University of Twente, 7522 NB Enschede, The Netherlands

**Keywords:** neutralizing nanobody, viral infections, prevention, treatment

## Abstract

Public health crises triggered by viral infections pose severe threats to individual health and disrupt global socioeconomic systems. Against the backdrop of global pandemics caused by highly infectious diseases such as COVID-19 and Ebola virus disease (EVD), the development of innovative prevention and treatment strategies has become a strategic priority in the field of biomedicine. Neutralizing antibodies, as biological agents, are increasingly recognized for their potential in infectious disease control. Among these, nanobodies (Nbs) derived from camelid heavy-chain antibodies exhibit remarkable technical advantages due to their unique structural features. Compared to traditional neutralizing antibodies, nanobodies offer significant cost-effectiveness in production and enable versatile administration routes (e.g., subcutaneous injection, oral delivery, or aerosol inhalation), making them particularly suitable for respiratory infection control and resource-limited settings. Furthermore, engineered modification strategies—including multivalent constructs, multi-epitope recognition designs, and fragment crystallizable (Fc) domain fusion—effectively enhance their neutralizing activity and suppress viral immune escape mechanisms. Breakthroughs have been achieved in combating pathogens such as the Ebola virus and SARS-CoV-2, with mechanisms involving the blockade of virus–host interactions, induction of viral particle disintegration, and enhancement of immune responses. This review comprehensively discusses the structural characteristics, high-throughput screening technologies, and engineering strategies of nanobodies, providing theoretical foundations for the development of novel antiviral therapeutics. These advances hold strategic significance for addressing emerging and re-emerging infectious diseases.

## 1. Introduction

Since the breakthrough development of hybridoma technology by Köhler and Milstein for the production of monoclonal antibodies, the application of monoclonal antibodies (mAbs), characterized by their uniform physicochemical properties and single biological activity, has become ubiquitous in immunoassays, medical diagnostics, and therapeutic interventions [1]. Lu et al. [2] reported that eight of the world’s top ten best-selling drugs in 2018 were antibody-based biologics, generating an annual revenue of 115.2 billion US dollars, with projections indicating that this market will reach USD 300 billion by 2030, consolidating their dominance in the biopharmaceutical industry.

The evolution of antibody therapeutics has exposed inherent limitations in conventional antibodies that constrain industry progress. First, their large molecular size (~150 kDa) limits their ability to effectively penetrate tumor stroma and various biological barriers. Second, the structural complexity of conventional antibodies necessitates expensive and labor-intensive production processes. To overcome these challenges, genetic engineering has facilitated antibody miniaturization, yielding fragments like antigen-binding fragments (Fabs) and single-chain variable fragments (scFvs). However, these derivatives often suffer from compromised stability and suboptimal antigen affinity [3]. A paradigm shift occurred when Hamers-Casterman et al. identified naturally occurring heavy-chain antibodies (HcAbs) in camelids, which lack light chains [4]. This discovery enabled the engineering of single-domain antibodies (sdAbs), commonly referred to as nanobodies (Nbs), derived exclusively from the variable heavy-chain (VHH) domain. Nbs exhibit superior biophysical characteristics, including a low molecular weight (12–15 kDa), enhanced tissue penetration, exceptional stability, high solubility, and unique epitope recognition capabilities, sparking widespread interest in biomedical applications [5].

In recent years, infectious diseases caused by viruses have posed a formidable global health threat due to their high transmissibility, rapid mutability, and severe impact on public health and the economy. The emergence of viruses such as the novel coronavirus (COVID-19) [6], influenza [7], and novel zoonotic diseases [8] highlights the danger of viruses capable of rapid adaptation, immune evasion, and resistance to therapeutic interventions. These challenges emphasize the critical need for effective prevention and treatment strategies. Neutralizing antibodies (NAbs) serve as an important tool in the response to viral infections, being capable of directly blocking viral entry into host cells and mitigating disease progression [9,10]. Given these properties, NAbs hold great promise as potent therapeutics against both established and emerging viral threats [11]. Among these, Nbs exhibit unique advantages as neutralizing agents due to their inherent properties [12]. Their ability to access conformationally obscured or recessed viral epitopes, typically unreachable by conventional antibodies, confers potent neutralization potential. Additionally, Nbs maintain structural integrity and functional activity under a wide range of stress conditions, making them ideally suited for use in resource-limited environments [13]. Moreover, Nbs are cost-effective to produce in microbial systems, facilitating large-scale manufacturing and rapid deployment during epidemics. The above inherent attributes make neutralizing Nbs promising drug candidates against existing and emerging viral threats.

With the growing research on Nbs in combating viral infections, several reviews have summarized their use in this field [12,14,15]. However, these reviews have primarily focused on specific viruses such as HIV [14] and SARS-CoV-2 [12,15]. To provide a comprehensive overview of Nbs in antiviral therapy, this review systematically analyzes their applications across a broad spectrum of viral pathogens, aiming to highlight their versatility and potential in addressing diverse viral threats. This review is based on the Web of Science database, which retrieves academic papers published over the past decade using keywords such as “neutralizing Nbs”, ”antiviral Nbs”, and “virus-specific Nbs”. It comprehensively summarizes the structural and functional characteristics of Nbs, screening methodologies, engineering strategies, and their specific applications against various viral infections. Furthermore, the advantages and limitations of neutralizing Nbs are critically analyzed, and future perspectives on their development and clinical translation are explored. Drawing upon these collective insights, this review aims to inform future research directions and accelerate the clinical deployment of Nbs as next-generation antiviral agents.

## 2. Nb Characteristics

Nbs represent the smallest natural antigen-binding domains known to date. Their small size, high stability, ease of modification, and ability to recognize hidden epitopes make them highly valuable in biomedical applications. As neutralizing antibodies, Nbs hold unique advantages, including the ability to target hidden or conserved epitopes, making them particularly effective against rapidly evolving viruses [16]. Additionally, Nbs can be readily engineered into multivalent formats or fused with other therapeutic domains to enhance their neutralization potency, extend their half-life, and improve their pharmacokinetic properties [17,18]. Their high stability also allows for diverse administration routes, such as the aerosolized inhalation, which provides significant advantages in treating respiratory viral infections [19]. Collectively, the properties of Nbs make them ideal for the development of next-generation neutralizing antibodies.

### 2.1. Small Size and High Tissue Penetration

Nbs are a class of genetically engineered antibodies derived from heavy-chain antibodies or variable regions of neoantigen receptors, with a molecular weight of about 12-15 kDa, which is only one-tenth of that of conventional IgG antibodies, making them the smallest known functional antibody fragments (Figure 1A). The crystallographic studies reveal that Nbs exhibit an ellipsoid shape approximately 2.5 nm in diameter and 4.2 nm in length. The smaller size gives the Nbs superior tissue penetration ability, especially in the imaging and therapy of solid tumors. Pieterjan et al. [20] demonstrated in a mouse model that the monomeric Nbs achieved the maximum uptake within minutes of intravenous injection, whereas the contrasting monoclonal antibodies required 24 h to reach the maximum uptake. In addition, the rapid tissue permeability and short half-life of Nbs enhances their efficiency and safety in imaging, diagnostics, and certain therapeutic strategies, reducing the off-target toxicity in other tissues. Notably, multiple studies have confirmed that Nbs can cross the blood–brain barrier (BBB) for the imaging and treatment of a range of central nervous system (CNS) diseases, and this feature offers a potential solution to the challenge of delivering antibodies through the blood–brain barrier [21].

### 2.2. Simple Structure and High Modifiability

Nbs consist of single antigen-binding domains, which lack light chains and constant regions, distinguishing them from conventional antibodies (e.g., IgG). This simplified structure not only reduces the complexity of production and modification, but also increases their stability. Moreover, Nbs are encoded by a single gene, making them easy to modify by genetic engineering techniques. Ma et al. [22] successfully engineered a bispecific Nb dimer by engineering two Nbs targeting distinct epitopes, which broadly neutralize SARS-CoV-1 and SARS-CoV-2 variants, offering insights for the development of antibody drugs to neutralize mutation-prone viruses. In addition, some toxins, biomolecules, and reporter genes can also be conjugated to Nbs through genetic engineering or chemical modification. Liu et al. [23] catalyzed the targeted attachment of zanamivir, an anti-influenza virus drug, to an anti-mouse κ light-chain Nb by a sorting enzyme, and delivered the drug to the virus-infected cells via immunoglobulin binding. They demonstrated that the modified Nb effectively protected mice from influenza A and B infections. In addition, by taking advantage of the easy modification of Nbs, their affinity can be further enhanced by Nb modifications, such as multimerization strategies or affinity maturation.

### 2.3. Unique Antigen Recognition Mechanism

The complementary determining region (CDR) of an antibody consists of three complementary determining regions (CDR1, CDR2, and CDR3). Compared to conventional antibodies, Nbs generally have longer CDR1 and CDR3 regions, and some Nbs possess CDR3 regions containing up to 16–24 amino acid residues, significantly exceeding the 7–9 residues commonly found in conventional VH domains [24]. The elongated CDR3 region not only increases the antigen-binding diversity, but also enables the formation of additional disulfide bonds between the CDR3 and either CDR1 or framework region 2, stabilizing the CDR3 loop into a rigid “finger-like” structure capable of recognizing cryptic epitopes inaccessible to conventional antibodies, such as the protein clefts or the enzyme-catalytic sites, etc. Maeda et al. [25] developed a panel of Nbs that targeted the cryptic receptor-binding domain (RBD) clefts in SARS-CoV-2, which are difficult for conventional antibodies to recognize. Due to the relatively conserved protein clefts, which were less prone to mutation, this panel of Nbs was able to recognize the conserved hidden clefts of all SARS-CoV-2 spike variants including Omicron. In conclusion, the unique recognition pattern of Nbs enables them to recognize hidden epitopes, thus underscoring the inherent advantages of Nbs in developing neutralizing antibodies against highly mutable viruses.

### 2.4. High Solubility and Ease of Production

The heavy-chain variable region (VH) of Nbs consists of four relatively conserved framework regions (FR1, FR2, FR3, and FR4). Highly conserved hydrophobic amino acids in the FR2 region, such as Val37, Gly44, Leu45, and Trp47, interact with the light-chain variable region (VL) via hydrophobic interactions, facilitating whole antibody assembly. However, in the FR2 region of camel-derived Nbs, the absence of VL leads to the mutation of these hydrophobic residues into hydrophilic residues, significantly enhancing the water solubility of the Nbs (Figure 1B) [26]. These hydrophilic residues not only help maintain the conformational stability of monomeric Nbs, but also improve their production yield in various expression systems, including *E. coli* and yeast. Hennigan et al. [27] successfully achieved the scalable, robust, high-throughput expression, and purification of Nbs by using the autoDC REdox bacterial strain in a two-stage bioprocess, reaching a maximum expression level of 700 mg/L, significantly surpassing the production yield of conventional IgG antibodies. The excellent solubility and expression efficiency of Nbs support their broad applicability in therapeutic development.

### 2.5. Exceptional Stability and Tolerance

Conventional antibodies possess a complex quaternary structure that is prone to denaturation under harsh conditions, such as high temperatures, chemical denaturants, or proteases, leading to the loss of function. In contrast, Nbs exhibit remarkable stability. Firstly, Nbs demonstrate exceptional thermal stability, retaining their function even after prolonged exposure to high temperatures. This thermotolerance is attributed to their strong refolding capacity following denaturation, providing a significant advantage in resource-limited settings where refrigeration and infrastructure are scarce. Secondly, the presence of disulfide contributes to improving the structural stability of Nbs. Studies have shown that besides the existence of a pair of conserved disulfide bonds between the FR1 region (Cys23) and the FR3 region (Cys94), an additional disulfide bond may form between the CDR3 region and CDR1 region, which not only stabilizes the CDR3 loop, but also further reinforces the overall structural integrity of Nbs. In addition, Nbs are also tolerant to proteases and extreme pH conditions. Fiil et al. [28] developed a bivalent Nb that could preserve its activity against enterotoxigenic *Escherichia coli* (*E. coli*) in simulated porcine gastric fluid environment (containing proteases and extreme pH). When orally delivered as a feed additive to weaned piglets, this Nb could effectively prevent the proliferation of F4^+^ enterotoxigenic *E. coli* in the intestinal tract. Given their exceptional stability, Nbs provide a new opportunity for the development of orally administered antibody therapeutics.

### 2.6. Low Immunogenicity and High Druggability

Immunogenicity is a critical factor affecting the development of protein-based drugs, and the intrinsic properties of Nbs confer upon them inherently low immunogenicity. Firstly, the small size of Nbs reduces the number of potential immunogenic epitopes. In addition, the high solubility of Nbs minimizes the likelihood of forming highly immunogenic aggregates. More importantly, most Nbs share a high degree of sequence identity with human IGHV3 family gene products, exhibiting up to 95% identity in the FR region sequences [29]. A review of 35 clinical trials of VHH drugs, involving over 1000 patients and healthy volunteers, demonstrated that Nbs exhibit minimal immunogenicity compared to fully humanized IgG antibodies, with neutralizing anti-drug antibodies detected in fewer than 3% of subjects [30]. In conclusion, the intrinsic properties of Nbs, along with their high homology to human antibodies, makes them good candidates for therapeutic applications, with the advantages of low immunogenicity and high druggability.

## 3. Systematic Comparison Between Nbs and Conventional Antibody Formats

While Nbs exhibit remarkable properties for antiviral applications, it is critical to contextualize their advantages and limitations within the broader landscape of existing antibody formats, including mAbs, Fabs, and scFvs. To systematically compare their molecular characteristics, therapeutic potential, and clinical limitations, we summarize the key differences in Table 1, focusing on five critical dimensions: structural complexity, tissue penetration, production scalability, stability under physiological conditions, and clinical translation success. This comparative analysis highlights how nanobodies may complement or surpass conventional formats in addressing specific antiviral challenges.

In summary, while conventional antibody formats (mAbs, Fabs, scFvs) have pioneered the field of antiviral biologics, their limitations in terms of structural rigidity, production complexity, and restricted tissue accessibility highlight the niche for Nbs. The modular nature of Nbs further enables engineering strategies (e.g., multivalency, half-life extension) to overcome the challenges observed in traditional antibody formats, positioning them as versatile candidates for next-generation antiviral therapeutics.

## 4. Screening Techniques for Neutralizing Nbs

The development of neutralizing Nbs relies on efficient and precise screening methods. Advanced screening techniques not only speed up the development process, but also optimize the functional properties of the Nbs, facilitating the identification of candidate molecules with high affinity and specificity. Currently, the most commonly used methods for screening neutralizing Nbs include the following:

### 4.1. Phage Display Technology

Phage display technology was first proposed by Smith [31] in 1985. Its fundamental principle is to insert DNA sequences encoding exogenous polypeptides or proteins into the specific loci within phage coat protein genes. Upon infecting *Escherichia coli*, the engineered phages express fusion proteins that integrate both the exogenous sequence and the coat protein on the surface of the phage, thereby ensuring the genotype–phenotype unity while maintaining the physiological functions and viability of phages. Winter [32] used phage display technology to successfully develop the first fully human antibody drug, Adalimumab, which greatly facilitated the application of phage display in the field of drug development, and ultimately earned Smith and Winter the 2018 Nobel Prize in Chemistry for their achievements [33].

Phage display technology is also one of the most widely used methods in the discovery of Nbs, providing a robust platform for the selection of antigen-specific Nbs through iterative biopanning cycles consisting of four key steps: antigen binding, washing, elution, and amplification. These selection rounds can lead to the targeted enrichment of high-affinity Nbs [34]. The primary advantages of phage display technology include its ability to construct large-scale libraries with low equipment requirements and its efficiency in isolating high-affinity binding compounds by monovalent display. However, this approach also has notable limitations, including the necessity for multiple selection rounds, a time-intensive screening process, and potential non-specific binding due to phage surface adhesiveness. With the in-depth research into phage display technology, several phage display systems have been developed, including the filamentous phage display system, λ phage display system, and T4 and T7 phage display systems. Among these, M13 filamentous phage and T7 phage display systems are the most widely used [35].

Liu et al. [36] used the framework region of the commercial Nbs targeting von Willebrand factor as a template and customized their complementary determining regions (CDRs). This approach facilitated the construction of a large-capacity synthetic phage display Nb library, leading to the successful screening of multiple Nb candidates with a high affinity for the RBDs of SARS-CoV-2. In vitro neutralization experiments showed that VHH60 was able to efficiently and extensively inhibit the host infection with SARS-CoV-2 and the key mutant strains. Meanwhile, the animal experiments demonstrated that VHH60 significantly extended the survival time and survival rate of infected mice, highlighting its therapeutic potential. This research not only offered efficient Nbs for the treatment of SARS-CoV-2 infection, but also provided a reference for the rapid generation of therapeutic Nbs based on phage display technology.

### 4.2. Yeast Display Technology

Yeast surface display (YSD) was invented by Maarten P. Schreuder in 1997, and its basic principle is to fuse exogenous protein genes with specific carrier protein genes, utilizing the intracellular protein transport mechanisms of yeast to express the fusion proteins and display them on the surface of the yeast cells [37]. Yeast cells possess eukaryotic post-translational modification mechanisms that aid in proper protein folding, making this system highly suitable for displaying eukaryotic proteins such as antibodies, receptors, enzymes, and antigens. Currently, the most commonly used yeast surface display systems include the lectin system, Flocculin system, and glycosylphosphatidylinositol (GPI) anchoring system, etc. Among them, the most widely used is the α-lectin system, which consists of two subunits, Aga1 and Aga2, connected by disulfide bonds and anchored to the cell wall via a GPI anchor at the C-terminus of Aga1 protein, while the displayed proteins can be fused to either the N- or C-terminus of the Aga2 subunit [38,39].

Yeast surface display technology is also one of the most widely used techniques in the development of Nbs. When combined with fluorescence-activated cell sorting (FACS), it can be used to obtain target-specific Nbs through high-throughput and rapid screening. The key advantage of this technology lies in its eukaryotic expression system, which supports proper protein folding and maintains Nb activity, making it well-suited for high-throughput screening. However, its primary limitation is the relatively small library size compared to phage display technology, and the requirement for specialized and expensive equipment [39].

Li et al. [40] utilized yeast surface display technology to isolate 593 Nb candidates capable of targeting the SARS-CoV-2 spike protein from immunized alpacas. In addition, they identified a unique panel of Nbs that were effective against all the other variants of concern (VOCs) tested, including Omicron subvariants BA.1, BA.2, BA.2.12.1, and BA.4/5, as well as SARS-CoV-1 and various sarbecoviruses from bats and pangolins, making them the broadest and most effective SARS-CoV-2-neutralizing Nbs currently available.

## 5. Functional Evaluation of Neutralizing Nbs

The functional evaluation of neutralizing Nbs is a crucial step in both drug development and clinical application. Currently, the assessment methodologies for viral neutralizing antibodies include competitive binding assay, structural analysis, in vitro viral neutralization assay, in vivo protective study, and the evaluation of antibody-dependent enhancement (ADE) effects.

### 5.1. In Vitro Test Evaluation

The in vitro evaluation of neutralizing antibodies includes the pseudovirus neutralization assay and plaque reduction neutralization test. Among them, the pseudovirus neutralization assay utilizes genetic engineering techniques to express specific viral surface proteins on the surface of pseudovirus particles, giving them receptor-binding activity while lacking pathogenic infectivity. This method is particularly suitable for highly pathogenic viruses which can reduce the safety risk of experiments and allow for large-scale neutralizing antibody screening in laboratories with lower biosafety levels. In addition, the plaque reduction neutralization assay measures virus-induced reduction in cellular plaques after the co-treatment of antibodies and viruses in cell cultures, which is considered to be the gold standard for detecting the activity of neutralizing antibodies, and is particularly suitable for evaluating the direct neutralizing effect of antibodies on viruses.

Rossotti et al. [41] screened 37 Nbs, specifically recognizing SARS-CoV-2 spike proteins from immunized camels using phage display technology. Epitope sorting experiments via SPR and sandwich ELISA confirmed that 17 of these Nbs targeted six distinct epitopes within the RBD. The neutralizing activity of these 17 Nbs against the RBD was further validated through an in vitro pseudotyped lentivirus neutralization assay (PVNA). These in vitro neutralization tests provided essential reference data for subsequent in vivo neutralization studies.

### 5.2. In Vivo Test Evaluation

Before neutralizing antibodies can be introduced into clinical trials, their in vivo prophylactic or therapeutic efficacy needs to be evaluated using suitable animal models. An ideal animal infection model should possess genetic stability, distinct pathological features post-infection, and preferably a capacity to simulate human physiological responses under similar conditions. Currently, the commonly used animal models include mice, rats, ferrets, and non-human primates.

Based on in vitro neutralization tests, Cornish et al. [42] evaluated the prophylactic and therapeutic efficacy of the Nb H6 trimer in a Syrian hamster model infected with SARS-CoV-2 Omicron BA.5. The Nb was administered intranasally either 2 or 24 h before viral attack or 24 h after infection. The study confirmed that neutralizing Nbs effectively prevented weight loss in infected animals and reduced viral persistence.

Han et al. [43] screened 20 Nbs specifically targeting the SARS-CoV-2 RBDs from immunized alpacas via phage display technology. Among them, three Nbs effectively blocked the interaction between RBD and the ACE2 receptor, exhibiting broad neutralization against multiple SARS-CoV-2 variants, including D614G, Alpha, Beta, Gamma, and Delta, and the Omicron sublineages BA.1, BA.2, BA.4, and BA.5. In a severe COVID-19 mouse model, the intranasal administration of the three neutralizing Nbs above could effectively protect the mice and reduce the viral loads in both the upper and lower respiratory tracts.

## 6. Modifications of Neutralizing Nbs

After preparing and obtaining neutralizing Nbs, modifications are essential to enhance their functional properties, therapeutic potential, and in vivo stability. Currently, as shown in Figure 2, the main modification strategies for optimizing neutralizing Nbs include affinity maturation of neutralizing Nbs, Fc functionalization of neutralizing Nbs, and multimerization strategies of neutralizing Nbs.

### 6.1. Affinity Maturation of Neutralizing Nbs

The affinity of neutralizing Nbs directly affects their binding ability to viral targets and their neutralizing potency. By increasing the affinity, the neutralization activity of Nbs can be significantly enhanced, thereby increasing the efficacy of antiviral treatment. Moreover, high-affinity neutralizing Nbs can achieve effective viral inhibition at lower doses, which not only reduces treatment costs but also minimizes potential side effects.

Wellner et al. [46] reported a study utilizing Autonomous Hypermutation yeast display (AHEAD) to continuously perform the in vivo directed evolution of neutralizing Nbs. This technique, based on the OrthoRep system, utilized error-prone DNA polymerases to introduce continuous high-frequency mutations into the Nb genes during yeast proliferation. Coupled with flow cytometric sorting, this method successfully enhanced the affinity of an anti-SARS-CoV-2 RBD Nb (RBD10) by up to 580-fold within just two weeks. The affinity-matured Nb was then fused with an IgG Fc domain and its neutralization capacity was evaluated via pseudovirus neutralization assays, revealing a remarkable 925-fold increase in its neutralizing potency. This study confirmed that the affinity of neutralizing Nbs could be significantly enhanced through in vivo continuous directed evolution.

### 6.2. Fc Functionalization of Neutralizing Nbs

The small molecular weight of Nbs (~15 kDa) allows for rapid renal filtration, resulting in their relatively short half-life in the body and the requirement for frequent administration. At the same time, the absence of Fc segments in Nbs leads to their lack of Fc-mediated effector functions, such as antibody-dependent cell-mediated cytotoxicity (ADCC) and complement-dependent cytotoxicity (CDC), thereby limiting their therapeutic applications. To overcome these limitations, it is common to fuse Nbs with Fc segments, which significantly prolong their circulation time in the bloodstream, reduce the frequency of administration, and confer Fc-dependent immune functions.

Schriek et al. [47] fused HIV-1-neutralizing Nbs (J3, 2E7, or 1F10) to the IgG1 Fc segment to generate a complete Nb-Fc antibody, which had a bivalent character and significantly enhanced the affinity for the HIV-1 virus. Furthermore, the cellular experiments demonstrated that all Nb-Fc fusions effectively activated NK cells, indicating their potential to induce ADCC-mediated cell death.

### 6.3. Multimerization Strategies of Neutralizing Nbs

Monovalent Nbs have limited binding sites, making them less effective against multiepitope viruses or highly variable antigens. The antiviral neutralization activity of Nbs can be enhanced by multimerization strategies (e.g., dimerization or trimerization) or multispecificity strategies (e.g., bispecific antibody), which not only improve their affinity but also extend their in vivo half-life.

Liu et al. [44] screened and identified two panels of Nbs (R14 and S43) with neutralizing activity against pan-SARS-CoV-2 and pan-sarbecoviruses. In order to further enhance their potency, the authors developed three multivalent Nbs: a bivalent Nb, trivalent Nb, and decavalent Nb. In vitro experiments showed that compared to bivalent and trivalent R14, decavalent R14 exhibited an up to 94,000-fold greater neutralization activity against the Omicron pseudoviruses. In addition, animal experiments showed that aerosolized trivalent and decavalent R14 showed longer half-lives in the lungs (7.2 and 28.1 h) compared to aerosolized monomeric R14 (5.1 h). In addition, the multispecific Nbs, due to their ability to recognize multiple epitopes, not only enhanced viral neutralization but also reduced the risk of viral mutation escape.

Zheng et al. [48] screened a set of Nbs targeting the Omicron BA.1 spike RBD from a natural Nb phage display library. Pseudovirus neutralization assays showed that the heterotrimeric 2F2E5 had a 100-fold higher neutralization efficiency than the homotrimeric F2-tri. Moreover, 2F2E5 also maintained potent neutralization activity against mutant strains BF.7, XBB.1.5, and XBB.1.16, highlighting its ability to prevent viral escape due to mutations.

## 7. Application of Nbs in the Prevention and Treatment of Viral Infection

### 7.1. Ebola Virus

EVD is an acute and highly contagious disease caused by the Ebolavirus genus of the Filoviridae family, mainly affecting humans and non-human primates, with a fatality rate of up to 90%. The Ebola virus was first discovered in 1976 by Peter Piot and colleagues in the Democratic Republic of Congo. To date, five species of Ebola virus have been identified, four of which can cause infection in humans [49]. Typically, EVD outbreaks originate from zoonotic transmission and subsequently spread through human-to-human transmission via direct contact with infected individuals or exposure to contaminated bodily fluids [50].

In 2020, the U.S. Food and Drug Administration (FDA) approved two monoclonal antibody (MAb) drugs against Ebola: Inmazeb^TM^ and Ebanga^TM^. Inmazeb^TM^ is a cocktail composed of three monoclonal antibodies, whereas Ebanga^TM^ consists of a single monoclonal antibody (MAb114). Both drugs have been effective in improving patient survival in the 2018 to 2020 Ebola outbreak in the Congo by binding to glycoproteins on the surface of the Ebola virus and preventing the virus from entering host cells [51]. The success of these monoclonal antibody drugs shows the potential of neutralizing antibodies in the treatment of Ebola virus infections.

In recent years, Nbs, as an emerging type of genetically engineered antibodies, have been regarded as potential antiviral therapeutic drugs due to their unique intrinsic properties as well as potent neutralization activity. Esmagambetov et al. [50] developed a set of Nbs specifically targeting the Ebola virus glycoprotein (EBOV GP) through alpaca immunization. Among them, aEv6 exhibited high binding affinity and strong neutralization activity in pseudovirus-based assays, holding potential as a neutralizing antibody drug. To enhance its pharmacokinetics and immunological properties, aEv6 was fused with the Fc fragment of human IgG1, generating the chimeric antibody aEv6-Fc. In an immuno-suppressed mouse model, aEv6-Fc showed a protective effect in a lethal recombinant vesicular stomatitis virus (rVSV-GP) infection model and achieved a protection rate of approximately 30% within 2 h of administration. This study suggested that aEv6 and its modified derivatives held promise as potential therapeutic tools against Ebola virus in the future.

### 7.2. H5N1 Influenza Virus

Highly Pathogenic Avian Influenza Virus subtype H5N1 (HPAI H5N1) poses a great zoonotic threat to global public health due to its strong cross-species transmission capability and high lethality. The pathogenic mechanism of the H5N1 virus is closely associated with the functions of its two surface proteins—haemagglutinin (HA) and neuraminidase (NA). Among them, HA promotes the combination and invasion of the virus to host cells, while NA is responsible for the release and dissemination. Although conventional antiviral drugs (e.g., oseltamivir and zanamivir) are effective in common influenza cases, their effects are limited in the face of the H5N1 virus with rapid replication and higher lethality [52,53].

One effective strategy to combat H5N1 is targeting its neuraminidase (NA), which plays a crucial role in viral replication. To explore this approach, Cardoso et al. [52] utilized phage display technology to isolate a set of Nbs from immunized alpacas. Among them, N1-VHH demonstrated strong inhibitory activity against NA in vitro, significantly reducing viral replication. Additionally, the bivalent N1-VHH-Fc exhibited 30- to 240-fold higher antiviral efficacy than the monovalent VHH, providing robust prophylactic protection in a mouse model, and remaining effective even in oseltamivir-resistant H5N1 variants.

Another study by Ibáñez et al. [53] showed that both monovalent Nb (H5—VHHm) and bivalent Nb (H5—VHHb) targeting the H5N1 haemagglutinin (HA) were effective not only in preventing infections 4, 24, or 48 h prior to the viral attack but also in significantly reducing viral load in the lungs and delaying disease progression after infection. In comparative tests, H5—VHHb demonstrated a superior neutralizing activity to H5—VHHm, and prophylactic treatment with only a 0.5 μg dose of H5-VHHb achieved lung viral titers below the limit of detection, reflecting its potential as a low-dose but high-efficiency prophylactic agent against H5N1.

In summary, these studies demonstrated that Nbs targeting key viral proteins not only exhibited potent inhibitory activity in vitro but also conferred significant protection in vivo, making them promising candidates for H5N1 antiviral therapy.

### 7.3. Hepatitis Virus (HAV/HBV/HCV/HEV)

Hepatitis viruses, including hepatitis A (HAV), hepatitis B (HBV), hepatitis C (HCV), and hepatitis E (HEV), are major causes of liver disease worldwide, each possessing distinct structural features and pathogenic mechanisms. HAV is a non-enveloped, single-stranded RNA virus primarily transmitted via the fecal–oral route, typically causing acute self-limiting hepatitis. HBV, an enveloped, double-stranded DNA virus, spreads through blood and sexual contact, with infections potentially progressing to chronic liver disease and even liver cancer. HCV, an enveloped, single-stranded RNA virus, is mainly transmitted via blood, with a high likelihood of chronic infection following acute exposure. HEV, a non-enveloped, single-stranded RNA virus, is mainly transmitted through contaminated water and usually causes acute hepatitis, posing a heightened risk of severe complications in pregnant women [54].

With advancements in hepatitis virus research, scientists are exploring more targeted antiviral strategies. For HBV research, Wang et al. [54] engineered an Nb 125s targeting the C-terminus region of hepatitis B virus surface antigen (HBsAg). When fused with an Fc fragment to form 125s-hFc, this Nb significantly reduced serum HBsAg levels, promoted immune responses, and modulated the expression of co-inhibitory molecules, offering a potential therapeutic approach for chronic HBV infection. For HCV prevention and treatment, Tarr et al. [55] utilized phage display technology to isolate four Nbs from HCV E2 glycoprotein-immunized alpacas. Among them, D03 was able to interfere with the interaction between E2 and CD81, thereby inhibiting HCV transmission between cells and showing strong antiviral potential.

In HEV prevention and treatment research, Chen et al. [56] used phage display technology to screen two Nbs, nb14 and nb53, which bind specifically to the HEV-associated protein sp239 from immunized camels. Neutralization capacity assays showed that nb14 and nb53 effectively blocked interactions between sp239 and HepG2 cells, which significantly reduced the copy number of intracellular HEV RNA and the viral infection rates, thereby suggesting new therapeutic approaches.

While previous studies have focused on monovalent Nbs for HEV, Wang et al. [57] took a step further by developing both monovalent (g3-p239-Nb55) and multivalent Nbs (fenobody-55) from HEV p239 protein-immunized camels, aiming to enhance antiviral efficacy. The results showed that both Nbs were able to target the HEV coat protein, resulting in effective neutralization. Notably, the multivalent fenobody-55 exhibited superior properties by self-assembling into a nanocage, which significantly enhanced its antigen-binding affinity (KD = 6.10 × 10^−9^ M, approximately 20-fold higher than that of g3-p239-Nb55) and prolonged its half-life to 267.5 min (approximately 9-fold longer than that of g3-p239-Nb55). These improvements significantly enhanced its ability to block viral attachment and infection.

### 7.4. Human Immunodeficiency Virus (HIV)

Human Immunodeficiency Virus (HIV), the etiological agent of Acquired Immunodeficiency Syndrome (AIDS), remains a major global public health concern due to its high transmission potential and ability to evade the host immune system. The viral structure of HIV is complex, consisting of a host-derived lipid bilayer envelope, virus-specific glycoproteins (e.g., gp120 and gp41), and a viral core, which contains a positive-sense RNA genome, etc. HIV infects CD4+ T-lymphocytes by binding to their surface receptors, leading to immune system collapse and the eventual immunodeficiency of the host [58].

The high genetic variability of HIV and its ability to evade immune surveillance present major challenges for conventional antiviral therapies. Nbs, owing to their small molecular weight, high tissue penetration, and structural stability, have emerged as a promising new-generation anti-HIV therapeutic strategy. These Nbs can specifically target the key structures of HIV, thereby inhibiting viral entry and replication.

Targeting the CD4-binding site of HIV-1 has been a promising strategy for Nb development. Several studies have explored this approach, with J3 emerging as a key candidate. McCoy et al. [58] demonstrated that J3 targeted the CD4-binding site of HIV-1, effectively inhibited both T-cell-to-T-cell and macrophage-to-T-cell HIV-1 transmission, and significantly reduced viral dissemination. Later, Zhou et al. [59] conducted a cryo-electron microscopy (cryo-EM) analysis and revealed that J3 binds to the HIV-1 envelope trimer in a manner similar to the CD4 receptor, stabilizing the prefusion-closed conformation. These findings provided structural insights that explained J3′s broad neutralization potency.

More recently, Zhang et al. [60] engineered a bispecific antibody, CAP256.J3LS, by fusing the light chain of the monoclonal antibody CAP256V2LS with Nb J3. CAP256.J3LS simultaneously targeted both the V2-apex region and the CD4-binding site of HIV-1, demonstrating superior efficacy in blocking diverse HIV-1 subtypes, particularly the endemic Subtype C in southern African. This antibody represented one of the most potent and broadly neutralizing HIV antibodies developed to date.

### 7.5. Human Papillomavirus (HPV)

Human papillomavirus (HPV), first identified in the 1970s, is a small double-stranded DNA virus with a genome of approximately 8000 base pairs, which encodes six early proteins (E1, E2, E4, E5, E6, E7) and two late proteins (L1, L2). Among them, the E6 and E7 are critical oncogenic proteins in high-risk HPV types, especially HPV16 and HPV18, which are associated with about 70% of cervical cancer cases worldwide. In particular, E6 inhibits apoptosis by degrading the oncogenic protein p53, whereas E7 contributes to aberrant cell proliferation by binding to retinoblastoma protein (RB), and the synergistic effect of these two leads to cellular carcinogenesis [61,62].

Due to the highly oncogenic potential and widespread prevalent nature of HPV, developing effective diagnostic and therapeutic strategies is an urgent issue. Although vaccines have significantly reduced HPV infections and made obvious progress, effective treatment options remain limited for patients already infected, especially those with HPV-associated cancers.

Li et al. [61] screened four Nbs targeting HPV16 E7 proteins by phage display technology. Subsequently, the authors expressed Nb2 within HPV16-positive cells. In proliferation experiments, it was found that the intracellular expression of Nb2 in HPV16-positive cells reduced cellular proliferation, highlighting its potential as an intracellular therapeutic agent for HPV16-associated diseases.

In another study, Togtema et al. [62] used phage display technology to isolate three high-affinity Nbs (A05, 2A12, and 2A17) from an immunized llama. These Nbs restored the function of the tumor suppressor protein p53 by blocking the interaction between E6 and the cellular E3A ubiquitin-protein ligase E6AP, thereby significantly inducing apoptosis in infected cells at the cellular level, laying the foundation for potential therapeutic strategies against HPV16.

Additionally, Minaeian et al. [63] screened two Nbs (sm5 and sm8) by phage display technology as well, which specifically recognized the major capsid protein L1 of HPV. In in vitro assays, both Nbs exhibited potent neutralization activity against HPV16 pseudoviruses. qRT-PCR experiments showed that sm5 and sm8 achieved 75% and 60% neutralization of HPV—16 virus particles at a concentration of 50 nM—suggesting their potential role in the prevention and treatment of HPV16 infections.

In summary, Nbs not only provide a new strategy to inhibit HPV infection, but also establish a solid foundation for the future clinical treatment of HPV-associated cancers.

### 7.6. Influenza A Virus (IAV)

Influenza A virus (IAV) is a highly mutable RNA virus that is capable of triggering seasonal influenza outbreaks, posing a serious threat to global public health. IAV is a negative-stranded RNA virus of the Orthomyxoviridae family, whose genome consists of eight segments of RNA encoding 11 proteins. Among them, hemagglutinin (HA) and neuraminidase (NA) are the most important surface antigens responsible for viral attachment and release, respectively. The high mutation rate and genetic reassortment capability of IAV enable it to evade host immunity, gradually reducing the effectiveness of conventional vaccines and antiviral therapies [64].

Based on this background, neutralizing Nbs have emerged as a promising approach for combating IAV infections. Voronina et al. [65] used phage display technology to identify two Nbs that specifically recognize the distal stem (stalk) domain (SD) of IAV HA. Animal experiments confirmed their effectiveness in neutralizing H1N1 and H5N2 influenza virus infections in vivo.

For the avian influenza H7 subtype, Huang et al. [66] isolated the Nb Nb-Z77, targeting H7 haemagglutinin (AIV-H7-HA) through camel immunization and yeast two-hybrid screening. Using multimerization strategies, they engineered Nb-Z77-derived constructs (Nb-Z77-DiGS, Nb-Z77-TriGS, Nb-Z77-Fc, NbZ77-Foldon) with significantly enhanced neutralization potency, completely inhibiting the HA activity of the inactivated viruses. These findings provided a foundation for developing effective neutralizing antibody therapies against AIV H7 subtype infections.

Shcheblyakov et al. [67] screened three broad-spectrum Nbs (D9.2, E12.2, and D4.2) targeting the HA protein of the H3N2 subtype from immunized camels using phage display technology. These Nbs efficiently blocked viral adsorption and invasion by specifically binding to key HA regions. Notably, the Fc-fused D9.2 Nb (D9.2-Fc) provided 100% protection in a lethal H3N2 mouse model, highlighting its strong therapeutic potential as an antibody drug against H3N2 influenza infections. This study illustrated the development of Nbs targeting different influenza A virus (IAV) proteins and their mechanisms of action in inhibiting IAV infection, laying the groundwork for the development of neutralizing Nb drugs against IAV.

### 7.7. Middle East Respiratory Syndrome Coronavirus (MERS-CoV)

Middle East Respiratory Syndrome Coronavirus (MERS-CoV) is an RNA virus that infects humans and other mammals, causing severe respiratory disease. It enters host cells by binding its spike protein (S protein) to the dipeptidyl peptidase 4 (DPP4) receptor on the host cell surface. MERS-CoV is highly pathogenic due to its rapid mutation rate, cross-species transmission potential, and ability to trigger a strong inflammatory response after infection [68].

Recent advances in Nb research have led to significant breakthroughs in MERS-CoV treatment. Zhao et al. [69] developed an Nb NbMS10, targeting the receptor-binding domain (RBD) of the MERS-CoV S protein using yeast display technology. NbMS10 exhibited a high affinity for the RBD and broad cross-neutralization activity against multiple MERS-CoV variants in vitro. In animal models, the Fc-fused NbMS10 (NbMS10-Fc) provided 100% protection in MERS-CoV-infected mice, demonstrating its potential as a therapeutic agent.

Stalin Raj et al. [68] identified four neutralizing Nbs from a one-humped camel infected with MERS-CoV. These Nbs blocked viral attachment and invasion by binding to the RBD of the MERS-CoV S protein. To enhance stability and half-life, the researchers engineered a chimeric camel–human heavy-chain antibody, combining the robustness of camel Nbs with the prolonged circulation time of human antibodies. This hybrid antibody provided strong protection against MERS-CoV infection in mouse models, offering new insights into optimizing Nb-based antiviral therapies.

### 7.8. Norovirus

Norovirus (NoV) has been a major cause of acute non-bacterial gastroenteritis since its discovery in 1972. It is a highly infectious, single-stranded, positive-sense RNA virus that encodes the capsid protein VP1, which forms the viral particle and facilitates infection by binding to histo-blood group antigen (HBGA) receptors on host cells. NoV spreads mainly through contaminated food, water, and direct contact [70]. Due to its high genetic diversity and antigenic drift, vaccine development and diagnostics remain challenging, making Nb-based antiviral strategies a promising alternative.

Earlier research by Koromyslova et al. [71] identified two Nbs, Nano-85 and Nano-25, targeting the VP1 capsid protein of human norovirus. Notably, Nano-85 induced the disassembly of NoV particles, highlighting its potential as an antiviral agent. More recently, the same research team identified nine Nbs targeting two clinically significant NoV GII genotypes (GII.4 and GII.17). Among them, four Nbs (NB-76, NB-2, NB-7, and NB-45) were found to interact with the HBGA-binding pocket on the P structural domain of norovirus, completely blocking NoV-HBGA interactions and showing promise as therapeutic antibody candidates [72].

In addition, Salmen et al. [73] found that the pre-developed Nb M4 induced viral particle disintegration by altering the conformations of norovirus capsid proteins, thereby inhibiting viral replication. This study provided a molecular basis for the further development of Nb-based antiviral drugs against norovirus.

In conclusion, these studies demonstrated that Nbs could inhibit NoV infection through multiple mechanisms, including viral particle disintegration and blocking viral–host interactions. These findings provided theoretical and data support for the development of Nb-based drugs against norovirus infections.

### 7.9. Poliovirus

Poliovirus (PV), the primary causative agent of poliomyelitis (polio), is a highly contagious enterovirus primarily transmitted via the fecal–oral route. After entering the digestive tract, PV undergoes extensive replication in the intestines before spreading through the bloodstream to the central nervous system, where it damages lower motor neurons, leading to acute flaccid paralysis. Since its discovery, PV has remained a significant public health concern. As a member of the *Picornaviridae* family, PV contains a single-stranded, positive-sense RNA genome enclosed within a capsid composed of four structural proteins: VP1, VP2, VP3, and VP4. Among them, the receptor-binding site on the VP1 protein plays a crucial role in viral attachment and entry in host cells [74].

Although vaccination has made significant progress in PV prevention, there is still an infection risk caused by vaccine-derived poliovirus (VDPV). To address this issue, the scientific community is actively searching for alternative means of prevention and therapy, among which Nbs have emerged as promising antiviral candidates due to their unique structure and potent neutralization ability [75].

Rombaut and Hogle’s team utilized phage display technology to screen a panel of Nbs from immunized one-humped camels, identifying several candidates that effectively bound to the PV capsid proteins. Five of them exhibited strong neutralizing ability in vitro, with half-maximal effective concentration (EC50) values in the nanomolar range. Further investigations demonstrated that these neutralizing Nbs inhibited PV infection by preventing viral attachment to host cells and interfering with the viral uncoating process [74,75].

To further investigate the molecular mechanisms of PV neutralization by these Nbs, the researchers utilized cryo-electron microscopy (Cryo-EM) to analyze the structures of five Nb-PV complexes. Their findings revealed that the binding sites of these Nbs overlapped with the PV receptor-binding site. Notably, the Nb PVSS21E not only bound to this site but also induced a conformation change in the PV capsid, transforming PV into a non-infectious extended state. This structural alteration effectively prevented PV from attaching to host cells, thereby neutralizing its infectivity [76]. In conclusion, these studies provided a foundation for developing Nb-based therapeutics against PV.

### 7.10. Rabies Virus

Rabies virus (RV) is a single-stranded, negative-sense RNA virus of the *Rhabdoviridae* family, characterized by its exceptionally high fatality rate. RV is primarily transmitted to humans through the saliva of infected animals, leading to lethal encephalitis. The rabies virus glycoprotein (G protein) facilitates viral entry by interacting with the acetylcholine receptors on neuronal cells. Once inside the host, RV spreads rapidly through the nervous system, causing irreversible neurological damage and ultimately death [77,78]. Despite advances in rabies prevention through vaccination, effective treatment remains limited once clinical symptoms appear, as the disease is nearly always fatal. To address this challenge, researchers have explored Nb-based therapeutics as a potential strategy against RV.

Hultberg et al. [77] utilized phage display technology to isolate multiple RV-neutralizing Nbs from an immunized llama. This research demonstrated that both bivalent and biparatopic Nbs significantly enhanced the neutralization potency and cross-protection, effectively inhibiting RV infection in an in vitro neutrality assay. Building upon these findings, the team further validated the therapeutic effectiveness of these Nbs in animal models. In a mouse model, lethal doses of RV were administered intranasally, followed by the direct intracerebral or intraperitoneal injection of anti-RV Nbs 24 h post-infection. The results showed that these Nbs effectively improved the survival rates, highlighting their potential for early intervention in rabies treatment [78].

To explore additional therapeutic strategies, the team explored the synergistic effect of combining anti-RV Nbs with vaccines. The results showed that mice in the combined treatment group exhibited significantly improved survival outcomes compared to vaccine-only or Nb-only groups. This study further confirmed the efficacy of Nb-based therapies in rabies control and provided valuable support for future combined treatment regimens based on Nbs and vaccines [79].

### 7.11. Respiratory Syncytial Virus (RSV)

Respiratory syncytial virus (RSV) is a negative-sense single-stranded RNA virus belonging to the *Paramyxoviridae* family, primarily transmitted through the respiratory tract. The structural protein (F protein) and hemagglutinin–neuraminidase protein (G protein) of RSV are its major antigenic components, with the F protein playing a crucial role in viral entry and pathogenesis. Specifically, the F protein mediates the fusion of the viral and host cell membranes, enabling subsequent infection. RSV poses a significant health threat to infants, young children, the elderly, and immunocompromised individuals, often leading to severe lower respiratory tract infections such as bronchiolitis and pneumonia. Although monoclonal antibodies, such as Palivizumab, have been developed for RSV prevention and treatment, their therapeutic efficacy remains limited. Furthermore, due to the rapid mutation of the virus, researchers have begun to explore the potential of Nb-based therapies for RSV infections [80,81]. Given its conserved nature, the RSV F protein has emerged as a promising target for Nb development.

To address this need, Detalle et al. [80] developed a trivalent Nb, ALX-0171, that specifically bound to the RSV F protein and blocked viral–host membrane fusion, thereby inhibiting viral replication and transmission. Unlike conventional antibodies, ALX-0171 exhibited superior antiviral activity and could be administered via nebulization, allowing for direct action at the infection site. This targeted delivery enhanced its efficacy, particularly in the lungs. In animal studies, nebulized ALX-0171 significantly reduced RSV viral loads in the lungs and nasal cavities of infected cotton rats, highlighting its therapeutic potential for RSV infections.

Building on these findings, Broadbent et al. [81] further evaluated the clinical potential of ALX-0171 by comparing its antiviral efficacy with that of the monoclonal antibody Palivizumab. Using a well-differentiated primary pediatric bronchial epithelial cell (WD-PBEC) model of RSV infection, their findings confirmed that ALX-0171 effectively reduced RSV replication and infection rates. Notably, ALX-0171 exhibited superior viral suppression compared to Palivizumab, providing compelling evidence to support its clinical application.

Additionally, Rossey et al. [82] developed two single-domain Nbs, F-VHH-4 and F-VHH-L66, that specifically recognized the RSV F protein. These Nbs bound to the prefusion conformation of the F protein, blocking its conformational transition during the early stages of viral infection and thereby inhibiting virus–host cell fusion. Animal experiments demonstrated that both Nbs significantly reduced RSV replication in vivo. In conclusion, these studies provided an innovative direction for developing Nb-based therapeutic strategies against RSV.

### 7.12. Rotavirus A

Rotavirus A (RVA) is the leading causative agent of severe diarrhea in children, which is primarily transmitted via the fecal–oral route. RVA infections can lead to acute watery diarrhea, which in severe cases may lead to dehydration and even death [83]. RVA, a member of the Reoviridae family, is a double-stranded RNA virus, whose genome consists of 11 segments encoding 11 proteins. Among them, the VP6 protein is the major structural component of the RVA inner capsid, exhibiting high immunogenicity and strong structural conservation. These properties make VP6 a crucial target for developing universal diagnostic tools and neutralizing antibodies [84].

In recent years, Nb-based strategies have shown promising applications in RVA prevention and treatment. In this regard, Gómez-Sebastián et al. [85] successfully established a high-yield expression system of two RVA-specific Nbs (3B2 and 2KD1) in insect larvae, achieving production levels of up to 257 mg/L. Through oral administration in mice, they demonstrated that both 3B2 and 2KD1 significantly alleviated RVA-induced diarrhea, and exhibited a synergistic protective effect when administered together.

Building upon these findings, further studies aimed to validate the protective efficacy of Nbs in more physiologically relevant animal models. Vega et al. [86] conducted a neonatal germ-free pig model, which closely mimicked human infant physiology, to assess the efficacy of orally administered Nb 3B2 in preventing diarrhea caused by a common human RVA strain (G1P). Their results demonstrated that supplementing the daily diet with 3B2 twice a day for nine consecutive days provided complete protection against RVA-induced diarrhea.

In conclusion, these studies not only highlighted the feasibility of large-scale Nb production but also the practical advantages of oral Nb administration, paving the way for the development of cost-effective and scalable RVA-preventive therapeutics.

### 7.13. SARS-CoV-2

Since the global outbreak of COVID-19 in late 2019, SARS-CoV-2 has rapidly emerged as a major public health challenge. As a member of the *Coronaviridae* family, SARS-CoV-2 is an enveloped, positive-stranded RNA virus with a genome of approximately 30 kb, encoding four major structural proteins: spike (S) protein, membrane (M) protein, envelope (E) protein, and nucleocapsid (N) protein. Among them, the S protein plays a crucial role in viral entry by binding to the angiotensin-converting enzyme 2 (ACE2) receptor on host cells, facilitating viral invasion and subsequent infection [87,88]. Following host infection, an aberrant immune response leads to severe pulmonary damage and multi-organ dysfunction, particularly in critical cases presenting with acute respiratory distress syndrome (ARDS).

To combat SARS-CoV-2, extensive efforts have been dedicated to developing neutralizing Nbs with broad-spectrum antiviral activity. Chi et al. [89] screened a panel of RBD-specific Nbs from a synthetic phage display library and subsequently engineered a biparatopic nanobody, Nb1–Nb2, by fusing two monomers (Nb1 and Nb2) that bind to distinct, non-overlapping epitopes on the SARS-CoV-2 RBD. This biparatopic configuration not only expanded epitope coverage but also significantly enhanced antiviral potency. In lentivirus-based pseudovirus assays, Nb1–Nb2 demonstrated potent neutralization across multiple SARS-CoV-2 variants of concern (VOCs), including Alpha, Beta, Gamma, Delta, Lambda, Kappa, and Mu, with sub-nanomolar IC₅₀ values ranging from 0.003 to 0.0865 nM. These results highlighted Nb1–Nb2 as a promising and broadly effective antiviral candidate against SARS-CoV-2 variants.

Beyond variant-specific neutralization, research also focused on developing Nbs with cross-reactivity against multiple coronaviruses. Laroche et al. [90] utilized deep mutational engineering to develop a broad-spectrum neutralizing Nb VHH72, which exhibited cross-reactivity against both SARS-CoV-1 and SARS-CoV-2. By optimizing its binding sites, the engineered Nb9 displayed increased resistance to antigenic drift, highlighting its potential for combating emerging coronavirus variants.

To further explore multivalent engineering strategies to enhance the neutralization potency and in vivo efficacy of Nbs, Yao et al. [91] used phage display technology to screen and obtain Nb4, an Nb capable of neutralizing Omicron subvariants (e.g., BA.1.1, BA.2.3, BA.5, and XBB.1.5) by targeting a conserved epitope within the S protein RBD. By obstructing RBD-ACE2 receptor interactions, Nb4 effectively inhibited viral entry into host cells. To enhance its neutralization potency, the team further developed a trivalent Nb, Nb4-16t, through multivalent engineering technology. In live-virus neutralization assays, Nb4-16t demonstrated significantly improved potency, with an IC50 value of 0.4 μg/mL against the BA.1.1 variant, compared to 1.9 ± 0.9 μg/mL for the monomeric Nb4. The study also explored different administration routes and found that intranasal administration of Nb4-16t provided superior protection in mouse models compared to intraperitoneal injection. Notably, the trivalent Nb Nb4-16t completely blocked infection by Omicron subvariants (BA.1.1, BA.2.3, BA.5, and XBB.1.5) and significantly reduced pulmonary viral loads. Additionally, this study also revealed the unique neutralization mechanism of Nb4 by using single-particle cryo-electron microscopy (cryo-EM) analysis, showing that Nb4 bound to both “down” and “up” conformations of the RBD, stabilizing the “2-RBD up/1-RBD down” configuration of the S protein, where “down” RBD binding was more stable while “up” RBD binding exposed more binding sites. This dual-conformation binding strategy enhanced the neutralization potency of the multivalent Nbs by providing greater epitope accessibility and stability.

Collectively, these studies highlighted the remarkable potential of Nb-based therapeutics in neutralizing SARS-CoV-2, particularly Omicron and its emerging variants. The development of multivalent Nb formats, such as Nb4-16t, demonstrated a promising strategy for enhancing neutralization efficacy. Furthermore, the successful implementation of intranasal administration paved the way for non-invasive antiviral Nb therapies, offering a practical and effective approach to COVID-19 prevention and treatment.

### 7.14. Venezuelan Equine Encephalitis Virus

Venezuelan equine encephalitis virus (VEEV) is a highly neurotoxic RNA virus belonging to the Alpha-virus genus. Its genome consists of single-stranded, positive-sense RNA that encodes various structural and non-structural proteins. Among them, the envelope glycoproteins E1 and E2 play a crucial role in viral entry by binding to host cell receptors and mediating infection. VEEV is primarily transmitted via mosquito vectors, and upon host infection, it rapidly invades the central nervous system (CNS), leading to a spectrum of clinical manifestations ranging from mild fever to severe encephalitis. In critical cases, VEEV infection may cause irreversible neurological damage or fatal outcomes [92,93]. Due to its rapid transmission, potential for aerosolized spread, and high pathogenicity, VEEV is not only a naturally occurring re-emerging pathogen but also a potential biological warfare agent.

With no approved antiviral treatments available, Nb-based neutralization strategies have gained attention for their high specificity, stability, and affinity for viral proteins. Liu et al. [94] developed a panel of high-affinity neutralizing Nbs targeting the VEEV capsid protein by immunizing llamas with an inactivated VEEV equine vaccine, followed by phage display selection. This panel included V3A8, V2C3, V2B3, and V3A8f, which were hypothesized to neutralize VEEV by specifically binding to the E1 and E2 glycoproteins, thereby inhibiting viral attachment to host cells and interfering with subsequent infection processes. The neutralization efficacy of these Nbs against the VEEV TC-83 strain was assessed using the plaque reduction neutralization test (PRNT), revealing that V3A8 (PRNT50 = 0.16 ± 0.02 μg/mL) and V3A8f (PRNT50 = 0.22 ± 0.08 μg/mL) exhibited the highest potency.

Although the monovalent Nbs exhibited significant neutralizing activity, additional optimization strategies were investigated to further enhance their therapeutic efficacy. To this end, the research engineered a series of hetero-divalent Nb constructs by linking two single-domain antibodies via a flexible linker. This approach significantly improved the neutralization potency, as evidenced by a substantial decrease in PRNT50 values. Specifically, the bivalent Nb V3A8f-V2B3 exhibited a PRNT50 of 0.76 ± 0.16 ng/mL, marking a 48-fold increase in neutralization efficiency compared to the mixed monovalent antibodies. Similarly, V2C3-V3A8f demonstrated a remarkable 287-fold improvement, with a PRNT50 of 0.71 ± 0.24 ng/mL. These findings underscored the potential of bivalent Nb constructs in achieving superior antiviral efficacy through synergistic mechanisms, providing a valuable foundation for the development of next-generation therapeutics against VEEV.

To provide a concise overview of the diverse antiviral Nbs described in the aforementioned sections, Table 2 summarizes key information regarding their viral targets, origins, binding epitopes, binding affinities, neutralizing activities, and mechanisms of action.

## 8. Advantages and Disadvantages of Nbs as Neutralizing Antibodies

Nbs possess distinctive structural and functional features that confer significant advantages in their use as viral neutralizing antibodies. Their single-domain structure allows for extensive genetic modifications, such as multimerization or fusion with therapeutic domains, enhancing their neutralization efficacy and extending their half-life. Additionally, some Nbs possess an extended complementarity-determining region 3 (CDR3), enabling them to form a finger-like structure that facilitates binding to conserved or cryptic epitopes inaccessible to conventional antibodies. This characteristic is particularly advantageous for developing neutralizing antibody drugs against highly mutated viruses. Furthermore, Nbs exhibit excellent water solubility and high production efficiency, as they can be expressed in microbial systems such as *Escherichia coli* or yeast, allowing for large-scale applications and rapid responses during pandemics. Their exceptional stability under extreme conditions, including resistance to variations in pH, temperature, and denaturation, supports alternative administration routes such as oral or aerosolized delivery, which are especially conducive to the treatment of respiratory virus infections. Lastly, Nbs demonstrate low immunogenicity and high druggability due to their small molecular weight and high sequence homology with antibodies, reducing the likelihood of eliciting immune responses when used therapeutically.

Despite these advantages, there are also several challenges when Nbs are used as neutralizing antibodies. One major limitation is their short half-life in circulation, attributed to their small molecular weight, necessitating PEGylation or Fc-fusion modifications to prolong systemic retention. Moreover, unlike conventional antibodies, Nbs lack an Fc region, which restricts their ability to mediate antibody-dependent cellular cytotoxicity (ADCC) and complement activation, thereby limiting immune effector functions. Lastly, regulatory approval poses additional hurdles; as novel antibodies, Nb-based therapeutics may face more stringent scrutiny compared to well-established monoclonal antibodies.

## 9. Conclusions and Prospects

Nbs have emerged as a promising class of antiviral agents based on their unique structural advantages and engineering flexibility. Their ability to access conserved or cryptic epitopes and to be formatted into multivalent constructs enables the effective neutralization of diverse and rapidly evolving viral pathogens. Over the past decade, numerous Nbs have been developed against various viral pathogens, including respiratory syncytial virus (RSV), influenza virus, and SARS-CoV-2. The ease of genetic modifications enables the generation of multivalent and multispecific Nb constructs, which not only enhance the neutralization potency but also expand their applicability across multiple viral strains and variants. In addition, the cost-effective production of Nbs in microbial systems facilitates their rapid deployment during large-scale outbreaks and enhances their accessibility in resource-limited and medically deprived areas.

Looking ahead, continued advancements in delivery systems, such as aerosolized formulations for respiratory infections, will further expand the clinical utility of Nbs. Additionally, with the emergence of approved Nb-based therapeutics, their regulatory pathway is becoming increasingly standardized, paving the way for broader clinical applications. In the future, combining Nbs with conventional antibody therapies or antiviral drugs may offer synergistic effects, providing more effective treatment options against emerging and re-emerging viral threats.

## Figures and Tables

**Figure 1 microorganisms-13-01352-f001:**
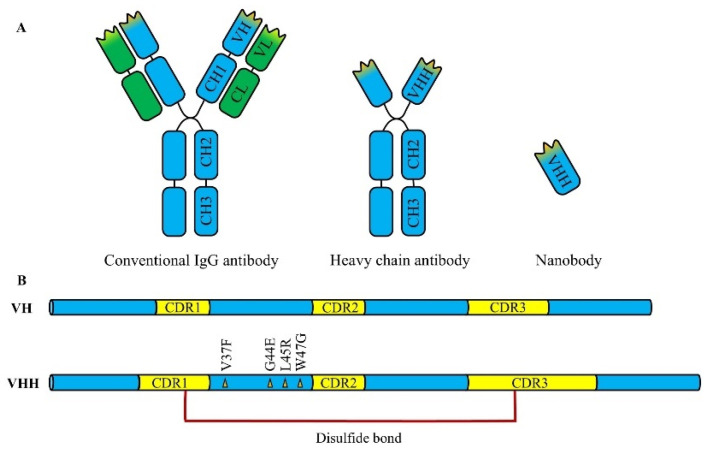
Comparison between a conventional IgG antibody, heavy-chain antibody (HcAb), and Nb. (**A**) Schematic illustration of a conventional IgG antibody (left), composed of two heavy chains and two light chains; a heavy-chain antibody (middle), lacking light chains and naturally occurring in camelids; and an Nb (right), representing a single variable domain (VHH) derived from HcAbs. (**B**) Comparative structural features of conventional VH and camelid-derived VHH domains. The VHH domain contains hallmark amino acid substitutions (e.g., V37F, G44E, L45R, W47G) that increase solubility and stability in the absence of a light chain. A unique disulfide bond between CDR1 and CDR3 further enhances its conformational stability.

**Figure 2 microorganisms-13-01352-f002:**
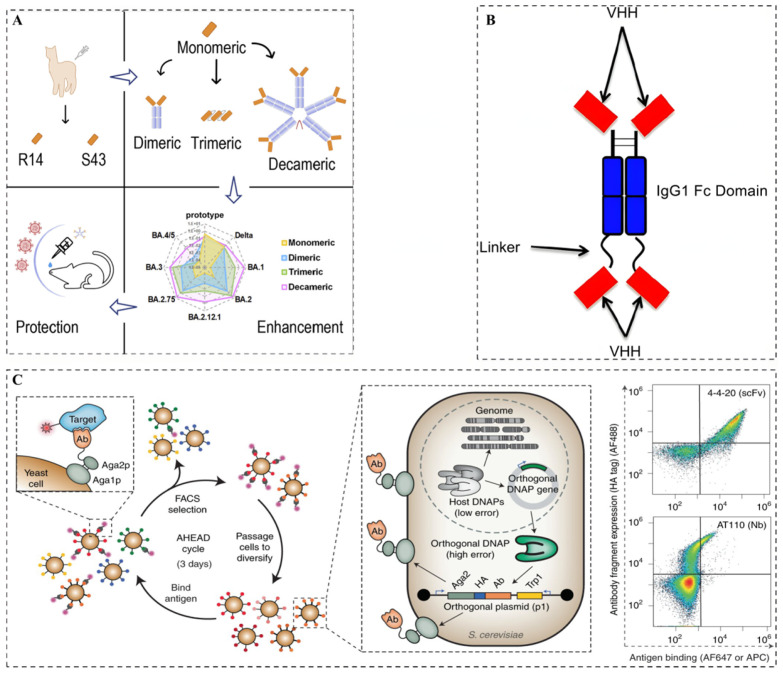
Three common neutralizing Nb engineering strategies. (**A**) Multimerization strategies of neutralizing Nbs [44], (**B**) Fc functionalization of neutralizing Nbs [45], (**C**) affinity maturation of neutralizing Nbs [46].

**Table 1 microorganisms-13-01352-t001:** A systematic comparison between Nbs and conventional antibody formats in antiviral applications.

Dimension	mAbs	Fabs	scFvs	Nbs
Molecular weight	~150 kDa	~50 kDa	~25 kDa	~15 kDa
Production system	Mammalian cells	Mammalian cells	Yeast/*E. coli*	Yeast/*E. coli*
Stability	+++	++	++	++++
Paratope diversity	++	++	++	+++
Tissue penetration	+	++	+++	++++
Gene modification	+	++	+++	++++
Clinical approval	+++	++	++	+

The quantity of “+” symbols demonstrates a positive correlation with the parameters.

**Table 2 microorganisms-13-01352-t002:** The application of Nbs in the prevention and treatment of viral infections.

Virus	Name	Origin	Binding Region	Affinity	Neutralizing Activity	Mechanism of Action	Ref.
Ebola	aEv6-Fc	Immunized Alpaca	RBD	N/A	N/A	Blocking viral attachment to host cell; enhancing immune-mediated viral clearance	[50]
H5N1	N1-5-VHHb	Immunized Alpaca	Neuraminidase (NA)	N/A	0.69 ± 0.231 nM	Inhibiting viral replication	[52]
	H5-VHHb	Immunized Alpaca	Hemagglutinin (HA)	N/A	N/A	Blocking viral attachment to host cell receptors	[53]
HBV	125s-hFc	Immunized Alpaca	C-terminus of HBsAg	N/A	N/A	Enhancing immune-mediated viral clearance	[54]
HCV	D03	Immunized Alpaca	CD81-Binding Site	N/A	1–10 μg/mL	Blocking viral budding and dissemination	[55]
HEV	nb14, b53	Immunized Camel	ORF2 Protein	N/A	N/A	Blocking viral attachment to host cell receptors	[56]
	g3-p239-Nb55	Immunized Camel	P239 Protein	0.299 nM	N/A	Blocking viral attachment to host cell receptors	[57]
	Fenobody-55	Immunized Camel	P239 Protein	6.10 nM	N/A	Blocking viral attachment to host cell receptors	[57]
HIV	J3	Immunized Llama	CD4-Binding Site	N/A	0.256 mg/mL	Blocking viral attachment to host cell receptors	[58,59]
	CAP256.J3LS	Immunized Llama	CD4-Binding Site and V2-apex Region	N/A	0.05 μg/mL	Blocking viral attachment to host cell receptors	[60]
HPV	Nb2	Immunized Camel	E7 Protein	N/A	N/A	Inhibiting viral replication	[61]
	A05	Immunized Llama	E6 Protein	122/398 nM	N/A	Inhibiting viral replication	[62]
	2A12	Immunized Llama	E6 Protein	21/93 nM	N/A	Inhibiting viral replication	[62]
	2A17	Immunized Llama	E6 Protein	40/59 nM	N/A	Inhibiting viral replication	[62]
	sm5, sm8	Immunized Camel	L1 Protein	N/A	N/A	Blocking viral attachment to host cell receptors	[63]
IAV	H1.2	Immunized Alpaca	HA Distal Stem domain	0.365 nM	N/A	Blocking viral attachment to host cell receptors	[65]
	G2.3	Immunized Alpaca	HA Distal Stem Domain	0.554 nM	N/A	Blocking viral attachment to host cell receptors	[65]
	Nb-Z77	Immunized Camel	Hemagglutinin (HA)	97.9 nM	30.89 nM	Blocking viral attachment to host cell receptors	[66]
	D9.2-Fc	Immunized Camel	Hemagglutinin (HA)	0.46 nM	N/A	Blocking viral attachment to host cell receptors	[67]
MERS-CoV	NbMS10	Immunized Alpaca	RBD	0.87nM	3.52 μg/mL	Blocking viral attachment to host cell receptors	[68]
Norovirus	Nano-85	Immunized Alpaca	VP1 Protein	3.5/7 nM	N/A	Promoting viral particle dissociation	[71]
	NB-45	Immunized Alpaca	VP2 Protein	101 ± 82 nM	0.46 μg/mL	Blocking viral attachment to host cell receptors	[72]
	M4	Immunized Llama	P-domain	<0.001 nM	53/379 ng/mL	Promoting viral particle dissociation	[72]
Poliovirus	PVSS21E	Immunized Camel	VP1 Protein	83.5 nM	84 nM	Promoting viral particle dissociation	[76]
RV	Rab-E8/H7	Immunized Llama	G Protein	N/A	0.14 nM	Enhancing immune-mediated viral clearance	[78]
RSV	ALX-0171	Immunized Llama	F Protein	0.113 nM	0.1 ± 0.07 nM	Blocking viral attachment to host cell receptors	[80]
	VHH-4, VHH-L66	Immunized Llama	F Protein	18/154 pM	0.1 nM	Blocking viral attachment to host cell receptors	[82]
RVA	3B2, 2KD1	Immunized Alpaca	VP6 Protein	N/A	0.06/0.24 μg/mL	Blocking viral attachment to host cell receptors	[85,86]
SARS-CoV-2	Nb1-Nb2	Synthetic Phage Library	RBD	<0.001 nM	0.003–0.0865 nM	Blocking viral attachment to host cell receptors	[89]
	Nb9	Immunized Llama	RBD	1.2/60 nM	N/A	Blocking viral attachment to host cell receptors	[90]
	Nb4	Synthetic Phage Library	RBD	8 nM	1.9 ± 0.9 μg/mL	Blocking viral attachment to host cell receptors	[91]
	Nb4-16t	Synthetic Phage Library	RBD	2.2 nM	0.4–22 μg/mL	Blocking viral attachment to host cell receptors	[91]
VEEV	V3A8, V3A8f	Immunized Llama	E1/E2 Glycoproteins	N/A	0.16 ± 0.02 µg/mL	Blocking viral attachment to host cell receptors	[94]
	V3A8f-V2B3, V2C3-V3A8f	Immunized Llama	E1/E2 Glycoproteins	N/A	0.76 ± 0.16 ng/mL	Blocking viral attachment to host cell receptors	[94]

## Data Availability

No new data were created or analyzed in this study.

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
