# Peer review of "Research Progress on the Application of Neutralizing Nanobodies in the Prevention and Treatment of Viral Infections"

_microorganisms, 2025, doi:10.3390/microorganisms13061352_

Round 1
Reviewer 1 Report
Comments and Suggestions for Authors
Thanks to the editor of the Journal for the opportunity to review the manuscript entitled: "Research Progress on the Application of Neutralizing Nanobody in the Prevention and Treatment of Viral Infections" by Duan et al.
I have reviewed the manuscript carefully. It is a comprehensive and useful review. The manuscript is well-structured and timely. However, there are some areas that need improvement to make it clearer and more focused.
General Comments
- The manuscript is well-organized and informative.
- It covers a wide range of topics about nanobodies and viral infections.
- Some parts can be improved to reduce repetition and enhance clarity.
Major Comments
- A comparison with traditional antibody formats (monoclonal antibodies, scFvs, Fabs) would improve the depth of the review. Please discuss their success, limitations, and clinical translation.
- Some descriptions about nanobody features (small size, stability) are repeated in different sections. Please consolidate to avoid redundancy.
- Several sections repeat the advantages of nanobodies (small size, stability, cost-effectiveness). Please condense to avoid unnecessary repetition.
- Table 1 is informative but very dense. Please consider:
- Grouping viruses into categories (e.g., respiratory, hemorrhagic, enteric).
Specific Suggestions
Figures
- Improve the resolution and labeling of Figures 1A and 1 B.
- Expand the figure legend to explain key differences between IgG, HcAb, and Nb.
References
- Ensure consistent formatting of in-text citations (e.g., spacing before/after references).
- Check that all references are complete and accurate, especially references [47], [49], [53], [57], and [85].
Grammar and Style
- - Maintain consistent tense when describing previous studies (preferably past tense).
- Define abbreviations (e.g., ADE, Fc, RBD) at first use.
- Simplify long sentences to improve readability.
Author Response
Comments 1: A comparison with traditional antibody formats (monoclonal antibodies, scFvs, Fabs) would improve the depth of the review. Please discuss their success, limitations, and clinical translation.
Responses 1: We sincerely appreciate the reviewer's valuable suggestion. In response, we have added a new section entitled "Systematic Comparison between Nanobodies and Conventional Antibody Formats" (Section 3, Page 5) that provides an in-depth analysis through seven critical dimensions: molecular weight, production system, stability, paratope diversity, tissue penetration, gene modification, clinical approval.
Comments 2: Some descriptions about nanobody features (small size, stability) are repeated in different sections. Please consolidate to avoid redundancy.
Several sections repeat the advantages of nanobodies (small size, stability, cost-effectiveness). Please condense to avoid unnecessary repetition.
Responses 2: We sincerely appreciate your expert feedback on the textual redundancy issues. We have thoroughly reviewed the entire text and specifically pruned redundant descriptions regarding the properties and advantages of nanobodies in Sections 1, 2, and 9 to eliminate unnecessary repetition.
Comments 3: Table 1 is informative but very dense. Please consider: Grouping viruses into categories (e.g., respiratory, hemorrhagic, enteric).
Responses 3: We concur with your observation regarding the information density in Table 1. While we have implemented partial streamlining, we have refrained from further viral categorization in the table, as the field currently lacks a universally standardized taxonomic framework for viral classification.
Comments 4: Improve the resolution and labeling of Figures 1A and 1 B. Expand the figure legend to explain key differences between IgG, HcAb, and Nb.
Responses 4: We have revised Figure 1, enhanced its clarity, and incorporated additional details. Furthermore, we have expanded the figure legend to include a comparative analysis highlighting the key distinctions among IgG, HcAb, and Nb.
Comments 5: Ensure consistent formatting of in-text citations (e.g., spacing before/after references).Check that all references are complete and accurate, especially references [47], [49], [53], [57], and [85].
Responses 5: We have standardized the formatting of all in-text citations to ensure consistent spacing around reference brackets. Additionally, we have meticulously verified the completeness and accuracy of references 47, 49, 53, 57, 59, 60, 66, 69, 81, 85, 89, 90, including authors, publication years, journal names.
Comments 6: Maintain consistent tense when describing previous studies (preferably past tense). Define abbreviations (e.g., ADE, Fc, RBD) at first use. Simplify long sentences to improve readability.
Responses 6: We sincerely appreciate the reviewer's insightful suggestions. The following revisions have been implemented: All descriptions of previous studies now consistently use the past tense; All abbreviations (e.g., ADE, Fc, RBD, Nbs) are fully defined at first mention (e.g., "fragment crystallizable (Fc)"); Overly complex sentences have been restructured for clarity.
Reviewer 2 Report
Comments and Suggestions for Authors
The authors present a review on the structural characteristics, development technologies, and engineering strategies of neutralizing nanobodies. While the topic is timely and relevant, several important aspects must be addressed before the manuscript can be considered for publication in Microorganisms. The following points should be carefully revised:
(1) The manuscript lacks crucial methodological details regarding the literature search, such as the search terms used, the databases consulted, the time frame covered, and the inclusion/exclusion criteria. Without this information, the review is not reproducible or verifiable, which undermines its scientific rigor.
(2) The review is heavily text-based and would greatly benefit from additional visual elements. The inclusion of more graphical content—such as summary tables, schematic diagrams, and illustrative figures—would enhance readability and help to better convey the recent advances in the field.
(3) The references are not formatted in accordance with the MDPI style guidelines. A thorough revision is required to align the citations with the journal’s standards.
(4) At least three recent reviews have addressed similar topics (see DOI: 10.3390/vaccines7030077, 10.3389/fimmu.2021.690742, and 10.1021/acsptsci.3c00042). The authors must clearly contextualize their work within the existing literature and explicitly state the novel contributions or perspectives offered by their review. This is essential to justify its relevance and originality.
Author Response
Comments 1: The manuscript lacks crucial methodological details regarding the literature search, such as the search terms used, the databases consulted, the time frame covered, and the inclusion/exclusion criteria. Without this information, the review is not reproducible or verifiable, which undermines its scientific rigor.
Responses 1: We sincerely appreciate the reviewer's suggestion regarding methodological transparency. We have supplemented the literature search protocol in Section 1 of the revised manuscript (Section 1, Page 2).
Comments 2: The review is heavily text-based and would greatly benefit from additional visual elements. The inclusion of more graphical content—such as summary tables, schematic diagrams, and illustrative figures—would enhance readability and help to better convey the recent advances in the field.
Responses 2: We sincerely thank the reviewer's suggestion. In response, we have added Figure 2 and Table 1 to the revised manuscript. Figure 2 illustrates the main modification strategies for optimizing neutralizing Nbs include: affinity maturation of neutralizing Nbs, Fc functionalization of neutralizing Nbs, multimerization strategies of neutralizing Nbs (Section 6, Page 8). While Table 1 provides a systematic comparison between Nbs and conventional antibody formats (mAbs, Fabs and scFvs) in antiviral applications across seven critical properties (Section 3, Page 5).
Comments 3: The references are not formatted in accordance with the MDPI style guidelines. A thorough revision is required to align the citations with the journal's standards.
Responses 3: We sincerely thank the reviewer for highlighting the formatting inconsistencies in the references. We have now thoroughly revised all citations to strictly adhere to MDPI style guidelines.
Comments 4: At least three recent reviews have addressed similar topics (see DOI: 10.3390/vaccines7030077, 10.3389/fimmu.2021.690742, and 10.1021/acsptsci.3c00042). The authors must clearly contextualize their work within the existing literature and explicitly state the novel contributions or perspectives offered by their review. This is essential to justify its relevance and originality.
Responses 4: We sincerely appreciate the reviewers' insightful feedback. With the increasing interest in neutralizing nanobodies (Nbs) for antiviral therapy has indeed prompted several recent reviews on their applications in viral neutralization, these works predominantly focus on narrow-spectrum case studies, such as HIV and SARS-CoV-2. To provide a comprehensive overview of Nbs in antiviral therapy, this review systematically analyzes their applications across a broad spectrum of viral pathogens, aiming to highlight their versatility and potential in addressing diverse viral threats.
Round 2
Reviewer 1 Report
Comments and Suggestions for Authors
Thank you to the authors for carefully revising the manuscript. The revisions have addressed the previous concerns effectively, and the updated version has improved in clarity and scientific rigor. I find the manuscript suitable for publication in its current form.